
# The role of history in and for climate science
# Social context and oral accounts

Hans von Storch[1,2]

[1]Institute of Coastal Systems, Helmholtz Zentrum Hereon, Geesthacht, Germany
[2]Meteorological Institute, Hamburg University, Hamburg, Germany

**Correspondence:** Hans von Storch (hvonstorch@web.de)

**Abstract.** The history of ideas is a key ingredient for understanding the knowledge generated and maintained by any science. In the case of western climate science this history is long has undergone various significant changes - beginning with climatic determinism, to physics of the atmosphere and oceans to a determinant of climate policies in recent times. These ideas went along with significant societal perceptions of climate, of "us" and "them", and using climate change as a tool to govern people.

As a natural science field, also climate science invests relatively little on the history of ideas, forgetting significant past personalities as well as past falsified concepts. In order to keep history alive, to "put people behind the science", in option is to resort to the old method of "oral history", to interview contemporary personalities about what happened, but also how they, and their social milieu, perceived the dynamics and role of climate. To do so, the author has archived many interviews with both matured climate scientists at the end of their career, but also with younger scholars.



# 1 Introduction

Today's discussion about the climate is dominated by concerns about the currently accelerated anthropogenic climate change. While in earlier decades, significant resistance prevailed not only in the public but also among some scientists, this "skepticims" has largely waned. The view that climate is changing to a less benevolent state, if not directly adverse to human life, has taken firmly the western public. Now, each extreme event, such as typhoons or strong rainfall events, are presented by media and

interest-led organizations as proof for ongoing anthropogenic climate change, and highly probable cascades of tipping points are predicted if the global mean air temperature increase crosses the magic $1.5°$ mark of the Paris agreement.

In this situation, a recourse on the fact that climate science is a social process (von Storch, 2023), and thus what is presented as strictly scientific may in part be less scientific but a manifestation of the changing "Zeitgeist", which is of course rooted in the history of public ideas. This is, where "history" becomes a significant approach to better understanding our science. It

turns out that understanding changes and extreme events as an expression of human influences on the climate is not uncommon (Section 2). Recourse to the explanation "man-made" is as not culturally new, at least in the West. The historical dimension also allows an analysis of the current discussion, particularly with regard to mitigation and adaptation strategies. An examples is the Swiss 19th Century Forest Police Law, briefly discussed in Section 2.

In section 3 the question is addressed, how we, rank-and-file climate scientists, can support the presence of past develop-

ments, of key personalities and past falsifications, in the collective memory of climate scientists. The author has done so by a number of interview series of matured climate scientists, who were present when climate science became climate change science, and of younger ones, who became part of climate science at a later time.

# 2 Perception of anthropogenic climate changes

The topic of "climate", i.e. the frequency of weather events and sequences, has been a ubiquituous object of perception and

interpretation in the human history (e.g. Glacken (1976)). Remarkably, the questions of the extent to which the climate has deliberately improved, for example by draining swamps, or to what extent the climate was unintentionally worsened by defor- estation (von Storch and Stehr, 2000, 2006), were often raised. The present knowledge, according to which man-made climate change would describe a new problem is therefore not correct. The notion of man-made climate change is part of our culturally constructed reality

It remains to be clarified to what extent the cultural construction influences the public acceptance of the scientific construc- tion "Climate change due to the emission of greenhouse gases". In fact, man-made climate change can not be explained by observations in everyday life as "man-made". The distinction between glacier retreats as a result of the end of the Little Ice Age or as a result of man-made warming can not be made by the layman; all you can see is that the glaciers are rapidly shrinking. This is consistent with the concept of human-caused climate change, but that it is mostly due to increased concentrations of

atmospheric greenhouse gases is an explanation of abstract science. The same is true of the ubiquitous claims that extreme weather events are indications of man-made climate change.



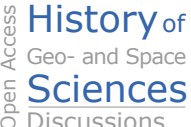

A general, detailed account, how people considered weather and climate, also from a practical view as well as significant thinkers and observers, is provided by Pfister and Wanner (2021) with a focus on Europe.

In this context, it is interesting to ask how people and societies in the past have explained why weather-related disasters
occurred. An example are the storm surges along the German North Sea coast (Jakubowski-Tiessen, 2004), which caused many deaths and extensive loss of land. Here was the explanation that the people had sinned and, thus, had conjured up their own fate. Another example (Stehr and von Storch, 1995) was the famine due to crop failures in the England of the 14th century, in which the authority of the time (the Church) acted with religious countermeasures against the climate catastrophe. The strategy with religious procedures to take action against bad weather were apparently fixed, at least in parts of Europe
institutionalized (Barriendos, 1997). in the 19th century, when modern meteorology began to emerge, the the scientific focus was on mean conditions, while extreme events were considered of little interest because "There is nothing to be learned from extraordinary events" (see Pfister and Wanner (2021), p.111).

A particularly interesting case is that of the Swiss Forest Police Act in the 1860's (Pfister and Brändli, 1999): In this case, an increased frequency of regional floods in Switzerland has been misrepresented as novel, that is, was perceived as having never
before happened. The historical records at that time were insufficient, so that earlier periods with intense flood episodes had been forgotten. The claim of the novelty of the phenomenon made a novel causative mechanism necessary - and was offered by the then still new forest science (Grove, 1975, e.g.): The excessive logging of trees in the high mountains. This should change the precipitation regimes and thus to more frequent flooding. As a result of this statement the Swiss forest police law was introduced nationwide, which forbid the harmful logging in the high mountains. There are certainly many good reasons
for preserving the forests in the high mountains, and in that respect the law was a very useful one, but the climatic justification with the risk of regional flooding was wrong.

At the end of the 19th century the practical impacts and needed countermeasures ´of the perceived climate change were widely discussed (as sketched in the introductory analysis of Brückner's dissertation (Brückner (1890); Stehr and von Storch (2000)). Already then the need was recognized to discriminate between anthropogenic influence and purely natural variability.
Parliamentary commissions in Prussia, Italy and Russia debated how to deal with the prospects of a changing climate (which usually went in the direction of reforestation).

It seems to me that reflecting on historical responses and perceptions can help to avoid repeating errors of the past, when discussing the present situation - namely to prematurely attributing causes to rare but possibly still natural events, and to inefficient decisions - while not to claim from such cases that the whole concept of anthropogenic climate change is a hoax.
This will lead to a more rational assessment of the risks and opportunities.



## 3 Interviewing Climate Scientists

Doing interviews with participants of past developments is commonly referred to as "oral history", which the Oral History Association [1] explains as being "a field of study and a method of gathering, preserving and interpreting the voices and memories of people, communities, and participants in past events."

### 3.1 The scientific legacy of Dennis Bray

A series of in-depth, semi-structured interviews were taped and transcribed in the mid 1990s. The interviews were based on a loosely formed interview protocol, with the nature of the dialogue shaping the subsequent probes and questions". The motivation for the interviews was to understand the perception of leading scientists, in Germany as well as in the US, about the then-perspective of climate change. A total of 26 interviews were done. These interviews were done promising anonymity, so
that only the general summary is publicly available; also three interviews were published after suitable anonymization.[2]

A major conclusion drawn from the interviews in the mid 1990s was "While the data demonstrates that, for the most part, the risk of global climate change is a consensual product of scientific practice, the hazards associated with the event are determined to have a much closer affinity with the scientist's personal belief system. It is often these beliefs that come to play a role in the application of science to the public and political spheres." (Bray and von Storch, 1996)

Based on these interviews, five surveys among international scholars were constructed and conducted; the responses are archived[3]. A number of publications were written using these responses:

- The first paper (Bray and von Storch, 1999) was on climate science being in a post-normal phase, by examining the four criteria of post-normality, namely inherent uncertainty, high stakes, social values involved, and urgency of decisions.

- The frequent failure of scholars to discriminate the concepts of "predictions" and "projection" is the topic of Bray and
von Storch (2009). In contradiction to the IPCC terminology, 29% of the respondents understand projections as probable developments , whereas about 20% of the respondents considered a prediction as a mere possible development.

- In Bray (2010) the degree of agreement with the IPCC reports is examined: "Results also suggest rather than a single group proclaiming the IPCC does not represent consensus, there are now two groups, one claiming the IPCC overestimates (a group previously labeled skeptics, deniers, etc.) and a relatively new formation of a group (many of whom have
participated in the IPCC process) proclaiming that IPCC tends to underestimate some climate related phenomena."

- Bray and von Storch (2011) compared the assessment the quality of climate models by two groups - one by experts engaged in the IPCC, and a second of "physical climate scientist". The responses in 3 consecutive surveys were evaluated - it turns out that expert group has more confidence in the models compared to the second group. The experts perceive a progress in recent years, whereas the others have essentially unchanged opinions.

---

[1] https://oralhistory.org/about/do-oral-history/
[2] Records and transcripts are archived, but not yet open to the scientific community because of promises given to the interviewees.
[3] Concerning archiving, see below



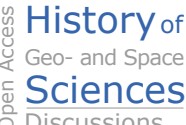

– Using the results of the 2013 survey concerning the norms of science, Bray and von Storch (2017) explores the climate scientists' subscription to CUDOS-norms of good scientific practice suggested by Robert Merton. The data suggests that while theses norms remain the overall guiding moral principles, they are not fully endorsed or present in the conduct of climate scientists: there is a tendency to withhold results until publication, there is the intention of maintaining property rights, there is external influence defining research and the tendency to assign the significance of authored work according

to the status of the author rather than content of the paper.

– The last paper in this series was von Storch and Bray (2017). The change of opinions among international climate scientists, as documented in the five surveys from 1996 until 1015/16. There was a strong increase in agreeing that warming is real and not influenced by changing measuring and reporting practices. Also the confidence has strongly grown that this change is due to ongoing anthropogenic causes. At the same time, however, the confidence in the climate modeling

has hardly increased during 1995-2015. In particular there is significant reservation concerning the representation of rainfall and clouds. Thus, the growing assessment of ongoing climate change as containing a significant anthropogenic component is based on several arguments: modeling is just one line of evidence - others are likely paleoclimatic evidence and statistical analysis of the instrumental record.

### 3.2 "Eminent" scientists

The second series of interviews features 14 eminent scientists (see http://www.hvonstorch.de/klima/interview.htm#individuals). These persons were chosen by Hans von Storch on the basis of his personal acquaintance and perception of significance, intellectual generosity, honesty, and open-mindedness. They have been part of the development of the 20th century science in one or the other way; they are "Zeitzeugen", people behind the science. At the time of the interview, they were typically 75 years old - so that they could look back on career extending over 50 and more years - and were somewhat detached from the

daily practice and responsibility of leading scientists. All interviewees were male, simply because there are hardly females in the age cohort of climate scientists. Unfortunately, quite a few have passed away since.

In all cases, Hans von Storch had asked a second interviewer to join - a person who would know the interviewee in a different context. The interviews were prepared by an extended personal exchange, if possible, to determine the issues of interest. The first few interviews were taped, later the questioning and responding was done by e-mail. No checking if the responses were

"correct" - the idea was to get the perceptions of the interviewees, not the "truth", whatever that may be - but even after many years we were not informed about significant inaccuracies in the responses.

– The first interview was done with **Hans Hinzpeter**(1921-1999; Figure 1), the late head of the Meteorological Department of Hamburg University and of a division of the Max-Planck Institute of Meteorology. An account of Hans Hinzpeter is given by https://de.wikipedia.org/wiki/Hans_Hinzpeter

The interview: von Storch, H. and K. Fraedrich, 1996: Interview mit Prof. Hans Hinzpeter, *Eigenverlag MPI für Meteorologie, Hamburg*; in German; DOI: 10.13140/RG.2.2.23236.83847

Online in English, at Niels-Bohr Library and Archives of the Center for History of Physics, (35176).



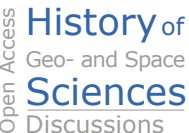

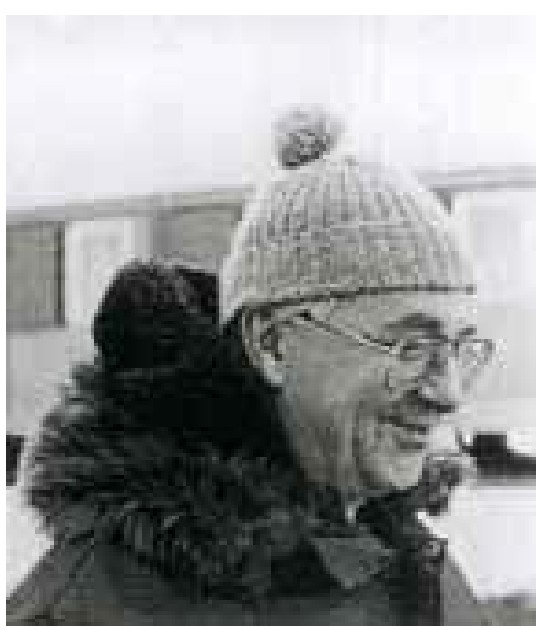

**Figure 1. Hans Hinzpeter**: The issues possibly of most interest was the sudden break of his career, when Eastern Germany blocked all travel from East Germany to the west in 1961, and the creation of the Max-Planck Institute of Meteorology in 1976.

- **Klaus Wyrtki** (1925-2013; Figure 2) was a oceanographer at the University of Hawai'i at Manoa in Honolulu. After having served in the German Navy during WWII, he moved to Indonesia to build a hydrographic service, and later to Honolulu, where he was a professor at the University of Hawai'i at Manoa. An account of Klaus Wyrtki is given at https://de.wikipedia.org/wiki/Klaus_Wyrtki

  The interview: von Storch, H., J. Sündermann and L. Magaard, 2000: Interview with Klaus Wyrtki. *GKSS Report 99/E/74*; 10.1007/0-387-33152-2_13; in English

  Also published in

  Historisch-Meereskundliches Jahrbuch 2000, Vol. 7, 49-94;

  Jochum M. and T. Murtugudde (eds.), 2006: Physical Oceanography. Developmennts since 1950, Springer Verlag, ISB 0-387-30261-1, 203-238,

  and online by the Niels-Bohr Library and Archives of the Center for History of Physics: (25031).

- **Reimar Lüst** (1923-2020; Figure 3) was a German astrophysicist, who became the powerful president of the Max-Planck-Society and later director general of the European Space Agency. More on him at https://en.wikipedia.org/wiki/-Reimar_Lüst

  The interview: von Storch, H., and K. Hasselmann, 2003: Interview mit Reimar Lüst. GKSS Report 2003/16, 39 pp, DOI: 10.13140/RG.2.2.22764.97928; in German

  English version at the Niels-Bohr Library and Archives of the Center for History of Physics (33761)



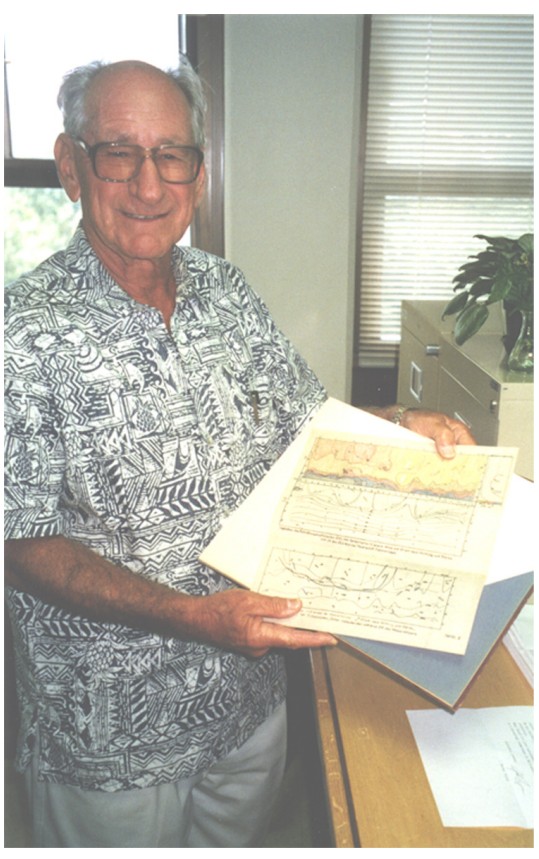

**Figure 2. Klaus Wyrtki**: A professor of oceanography at the University of Hawai'i at Manoa, who was best known for his studies on El Nino and the building of a sea level monitoring network across the Pacific Ocean.

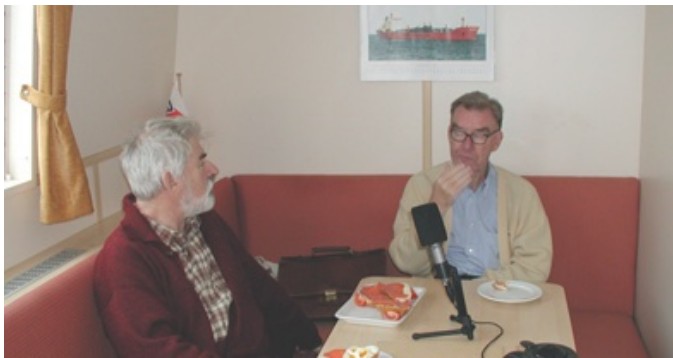

**Figure 3. Reimar Lüst** (right): As president of the Max-Planck Society, he decided to create a Max-Planck Institute of Meteorology in Hamburg, with **Klaus Hasselmann** (left) as director.





– **Harry van Loon** (1915-2021), a born Dane, went after WWII first to South Africa and then to the US, where he became a senior scientist at the National Center of Atmospheric Research. His main interests were atmospheric dynamics on the Southern Hemisphere, El Niño-Southern Oscillation and, in his later years, the link of climate and solar activity. Important advice to the young scholars were: do not become a full professor too early, and if using analyses, always try to validate the result with station data.

The interview: von Storch, H., G. Kiladis and R. Madden, 2005: Interview with Harry van Loon, *GKSS Report 2005/8*, DOI: 10.13140/RG.2.2.19409.53609; in English.

Also in the Niels-Bohr Library and Archives of the Center for History of Physics: https://www.aip.org/history-programs/niels-bohr-library/oral-histories/34462

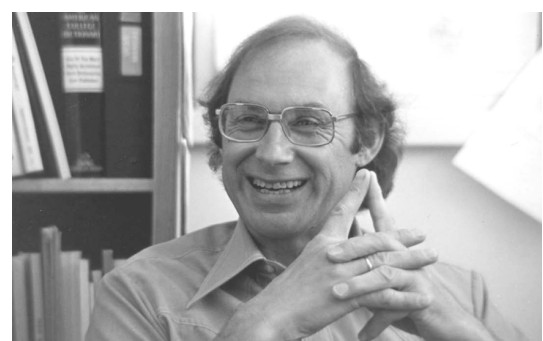

**Figure 4. Harry van Loon**: A senior scientist at NCAR, who traveled the world and paid visits to many scientific institutions, where he inspired myriads of young scholars. His interviews spells out many details about the early years of NCAR.

     – **Klaus Hasselmann** (1931- ; Figure 3), who was the founding director of the Max-Planck Institute of Meteorology and
of the German Climate Computing Center in Hamburg. His portfolio of interests ranged from ocean waves, climate dynamics, the interaction of climate and society to particle physics (von Storch (ed), 2022; von Storch and Heimbach, 2022). See also https://en.wikipedia.org/wiki/Klaus_Hasselmann

The interview: von Storch H., and D. Olbers, 2007: Interview with Klaus Hasselmann, *GKSS Report 2007/5*; 67 pp; // Also published online by the Niels-Bohr Library and Archives of the Center for History of Physics: 33645

– **Walter Munk** (1917-2021, Figure 5) was a towering oceanographer after WWII, a born Austrian. He moved to the US in the early 1930, and eventually ended up at Scripps Institution in La Jolla, California. His contributions refer to the angular momentum of Earth, and the length of day, the boundary currents and the Acoustic Thermometry of Ocean Climate. He worked with Harald Sverdrup to design a forecast system for ocean surface wave conditions, which was used at D-day, when allied troops landed in Normandy. He was also working closely with Klaus Hasselmann. See also
https://en.wikipedia.org/wiki/Walter_Munk

The interview: von Storch, H., and K. Hasselmann, 2010: Seventy Years of Exploration in Oceanography. A prolonged weekend discussion with Walter Munk. *Springer Publisher*, 137pp, DOI 10.1007/978-3-642-12087-9; open access



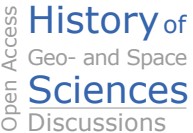

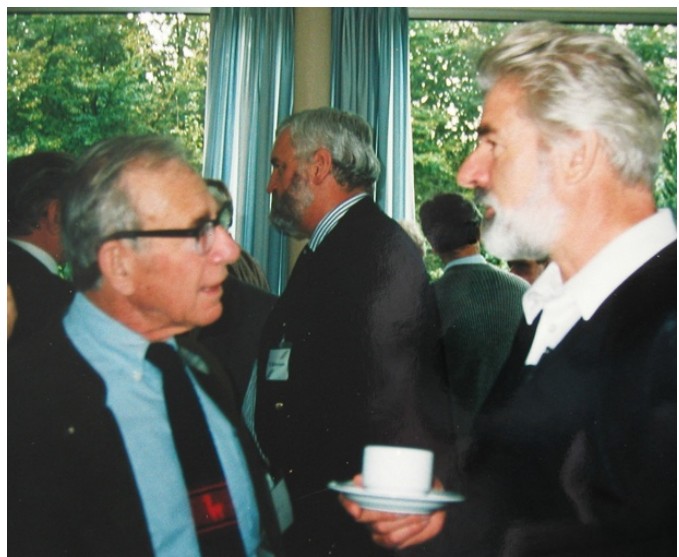

**Figure 5. Walter Munk** (left): a driving force in oceanography, which stood behind a number of significant findings and construction. He was advising the Max-Planck-Institute of Meteorology from the very beginning, and was a friend of Klaus Hasselmann (right). And he advised many US presidents.

The interview with Walter Munk was the last with personalities of age directly related to the Max-Planck Institute. After several years, the series was taken up again, this time mostly with other impressive, mostly somewhat younger scientists.


– **Hartmut Heinrich** (1952-) is possibly the most famous Earth scientist from Hamburg, next only to Klaus Hasselmann. He built his fame on a rock, which was recovered from 4000 m depth in the Northeast Atlantic; nobody paid any attention to it at that time, but Hartmut Heinrich, then a scientific nobody, took a closer look. He read the rock, asking how it would be possible to find an ice-rafted rock with traces of anoxia at that location. He developed the theory of what was later named by Wally Broecker "Heinrich event", when armadas of icebergs from ice sheets surrounding the North Atlantic


traveled southward, interrupting the oceanic conveyor belt, and indicated periods of short cold events.

Even though he lived and worked in Hamburg, he remained locally unknown, until the interview was done.

See also https://en.wikipedia.org/wiki/Hartmut_Heinrich

The interview: von Storch, H., and K. Emeis, 2017: Hartmut Heinrich - der unbekannte weltberühmte Klimaforscher aus Hamburg. *https://www.academia.edu/30989306/Hartmut_Heinrich _-_der_unbekannte_weltber%C3%BChmte _Klima-*


*forscher_aus_Hamburg._Ein_Interview*

English translation: doi 10.13140/RG.2.2.22909.15846

– **Jan Harff** (1943-; Figure 7) is a geologist from Eastern Germany, who was scientifically socialized in the German Democratic Republic. Before 1989 he was not allowed to visit Western countries , or to sail "westward" and concentrated his scientific activities on modeling of sedimentary basins. Thus he turned into marine geologist only after the fall of the



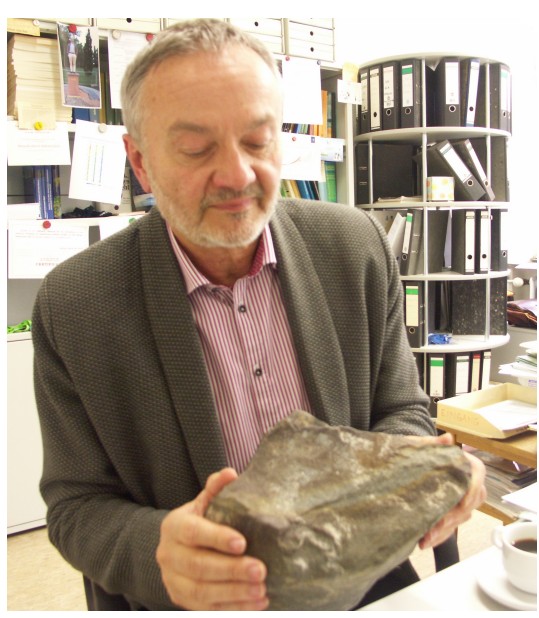

**Figure 6. Hartmut Heinrich** with "his stone" lifted from 4000 m depth in the Northeast Atlantic, from which he read the past sudden cold events which were later named "Heinrich events".

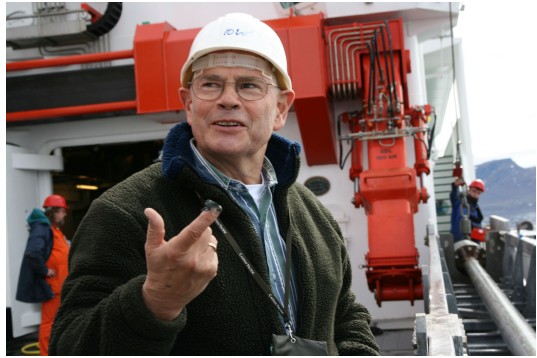

**Figure 7. Jan Harff** scientifically socialized in East Germany under communistic rule, limited in his travel to the east, prohibited from sailing the sea - and "liberated" from such limitations in 1990, which allowed him to catch up what he had missed so far - with significant work on the development of coasts all over the world, but with a certain focus on the Baltic Sea.



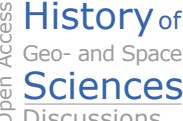

Berlin Wall, when he began something like a second scientific life, with a focus on marginal seas and in particular the development of the coast of the Southern Baltic Sea, also in the context of climate change.

The interview: von Storch, H., und R. Dietrich, 2017: Jan Harff - zwischen Welten. *www.academia.edu/35456080/Jan_Harff_zwischen_Welten*

also Historisch-Meereskundliches Jahrbuch 23, 2019, 219-264.

The interview was done in German, but was translated to Polish, Chinese, Russian and English:

hvonstorch.de/klima/Media/interviews/Jan.Harff-english.pdf, harff-polska.pdf, harff.china.pdf, and harff.russia.pdf.

– **Christian Pfister** (1944 - ; Figure 8) was the first social scientist interviewed in the series. Christian Pfister received the first Eduard-Brückner Prize[4] in 2000, for his interdisciplinary achievements climate science.He was professor for economic, social and environmental history at the University of Bern, Switzerland. He studied both, the history of
climate in Central Europe since about 1000 years, but also the history of perceptions of climate events.

See also https://en.wikipedia.org/wiki/Christian_Pfister_(Swiss_historian)

The interview: von Storch, H., and H. Wanner, 2019: Christian Pfister - ein Historiker, der Grenzen überwindet. *https://www.academia.edu/38311144/Christian_Pfister__ein_Historiker_der_Grenzen_%C3%BCberwindet*

In English: http://www.oeschger.unibe.ch/about_us/news/ interview_christian_pfister/index_eng.html

– **Jürgen Sündermann** (1938 - ; Figure 9) trained as a mathematician, he is a theoretical oceanographer at Hamburg University, interested in marginal seas, specifically the North Sea, and their numerical modeling, as well as the effect of tidal friction on the rotation of Earth and Moon.

von Storch H., und H. Langenberg, 2019: Interview mit Jürgen Sündermannm *HZG Bericht* 2019-1

DOI: 10.13140/RG.2.2.12015.48807

in Chinese: Media/interviews/suendermann.china.pdf https://www.academia.edu/38395308/

– **Anders Omstedt** (1949 -;Figure 10) is regional oceanographer of the University of Göteborg, Sweden, with keen interest in modeling the regional earth system of the Baltic Sea Region and in issues of philosophy of science and of the oceans. In about 30 years he was instrumental in governing the research program, named first BALTEX and later Baltic Earth.

The interview: von Storch, H., and M. Reckermann, 2020: Anders Omstedt - 45 years of wandering - from processes
to systems, through outer and inner seas. *International Baltic Earth Secretariat Publication* No. 17, February 2020, https://baltic.earth/interviews/

– **Geert Jan van Oldenborgh** (1961-2021; Figure 11) was located at the Royal Netherlands Meteorological Institute KNMI in de Bilt, and was a guest professor at Oxford University, were he joined and boosted the effort to build the
"event attribution" methodology. He was an open-minded physicist, who served the community and the public with robust climate diagnostics. The interview was done when he was already massively challenged by his disease.

---

[4]http://www.hvonstorch.de/klima/material/brueckner/brueckner-preis.htm



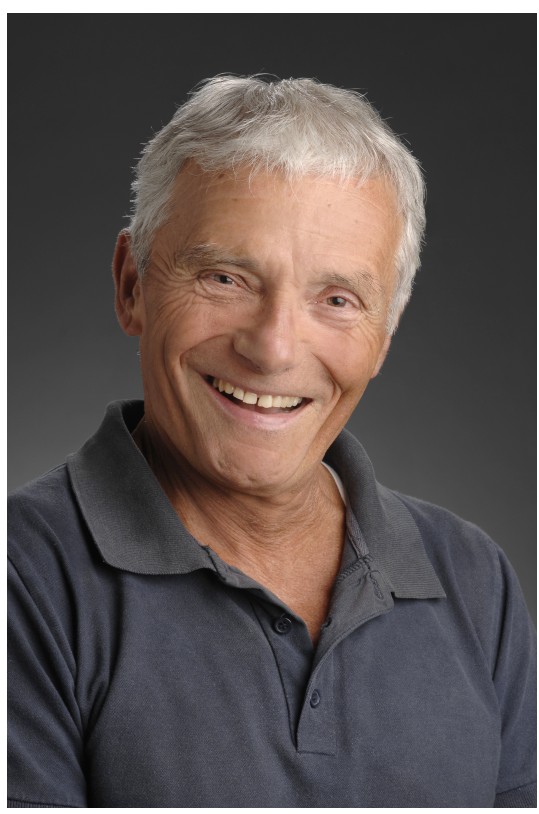

**Figure 8. Christian Pfister** - a rare species of a historian of environmental history, with focus in climate changes and perceptions thereof. A key publication is Pfister and Wanner (2021)

von Storch, H., and S. Philip, 2021: Geert Jan van Oldenborgh - approaching truth in a sea of noise and data deficiencies. *https://www.academia.edu/44946272/ Geert_Jan_van_Oldenborgh_approaching_truth_in_a_ sea_ of_noise_and_data_ deficiencies*

in Dutch: van Oldenborgh, G.J., S. Philip, H. von Storch, and M. Allan, 2021: Geert Jan van Oldenborgh - de waarheid benaderen in een zee van ruis en dataproblemen. *Meteorologica* 1-2021, 1-9

See also https://en.wikipedia.org/wiki/Geert_Jan_van_ Oldenborgh

– **Mike Hulme** (1960- ; Figure 12) is a geographer at the University of Cambridge, UK, whose interdisciplinary achievements were recognized by the 6th Eduard-Bruckner prize. As a founding director of the Tyndall Centre for Climate Change Research based at the University of East Anglia, he helped to analyse and neutralise the negative repercussions of the "climategate" affair.

See also https://en.wikipedia.org/wiki/Mike_Hulme

The interview: von Storch, H., and M. Claussen, 2021: Mike Hulme: The gentleman understanding climate beyond the




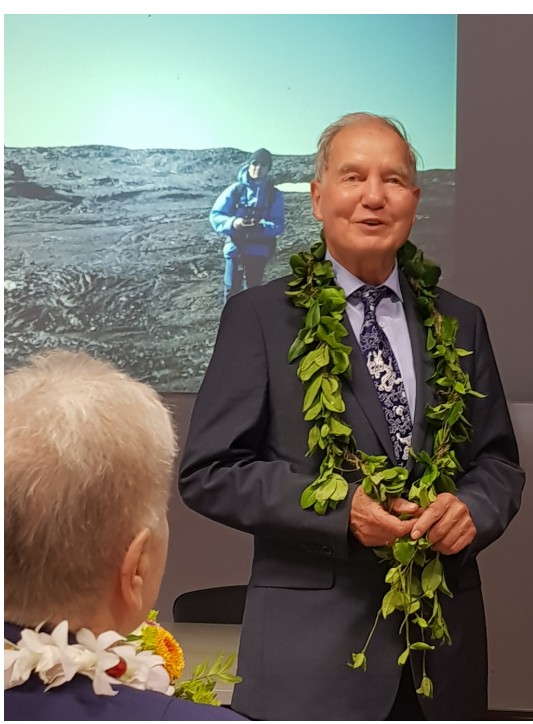

**Figure 9. Jürgen Sündermann** a regional oceanographer, who was one of the key architects of laying the foundations for a center of excellence of Climate Science in Hamburg

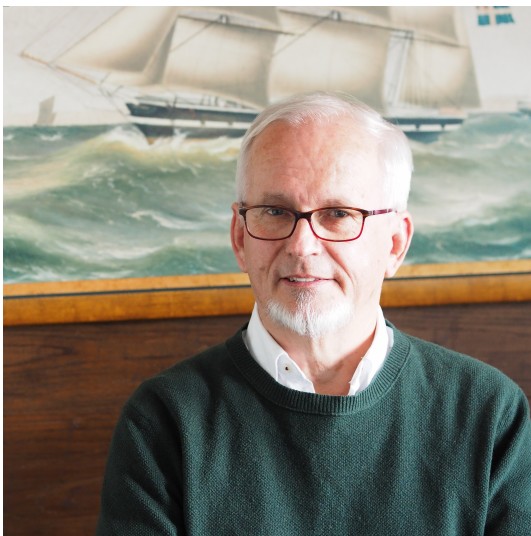

**Figure 10. Anders Omstedt** - also a regional oceanographer, but with a focus in the Baltic Sea - a key personality of the Baltic Seas region research, as brought together in the program BALTEX and Baltic Earth. Besides his competence about marine processes in the Baltic Sea, Omstedt has become known as philosopher of the sea (Omstedt, 2020).

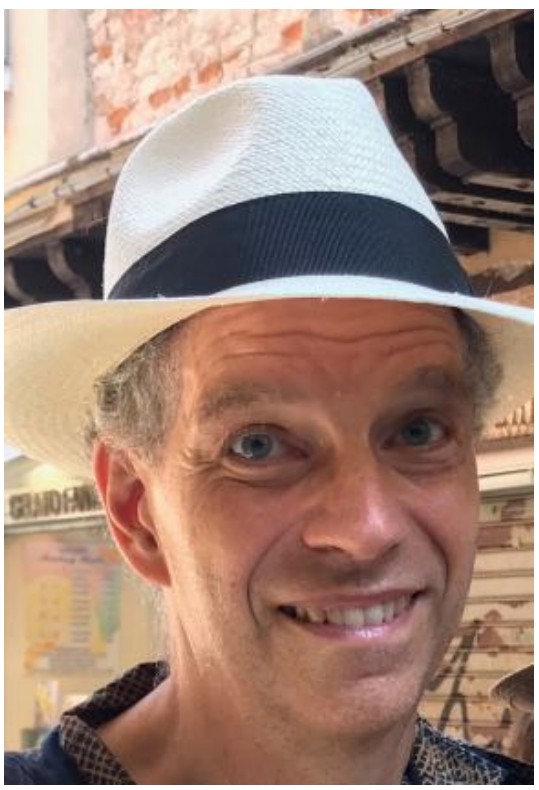

**Figure 11. Geert Jan van Oldenborgh** had a public and a scientific significance. In his public communication, he limiting himself to what could be demonstrated robustly, and he had a record of deconstructing various events, of which he found many as related to anthropogenic climate change but not all.

fascination of differential equations. *https://www.academia.edu/49459880/Mike_Hulme_ The_gentleman_ understand-*
*ing_climate_beyond_the _fascination_of_differential_equations*

- **Thomas J. Crowley** (1948-2014; Figure 13) was a geologist, whose main interest was on paleoclimatology, i.e., the geologic history of climate. A key legacy, which he left, was a joint book with Jerry North (Crowley and North, 1991). The interview was done, when he was already terminally ill.

  The interview: von Storch, H., H. Wanner and G. Hegerl, 2014: Scientific interview: Tom Crowley (1948-2014). *Dust*
22, 114-115 DOI: 10.13140/RG.2.2.21375.61605

## 3.3   Atmospheric Sciences Section of AGU

In the Newsletter of the Atmospheric Sciences Section of AGU Newsletter, Hans von Storch ran a series of short interviews with similar questions between July 2009 until March 2013. Matured atmospheric scientists were asked for their subjective views



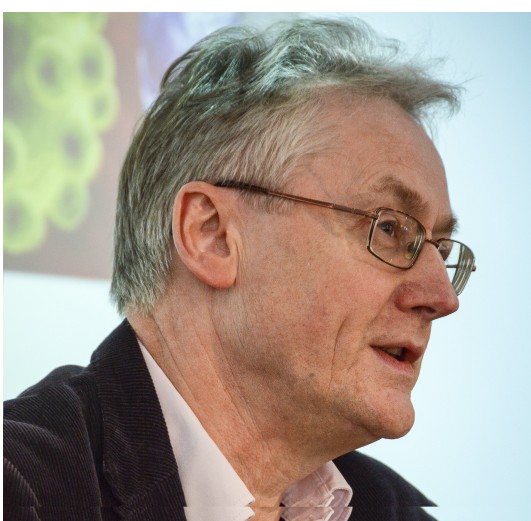

**Figure 12. Mike Hulme** is an industrious author of books, among them "Why we disagree about climate change" (Hulme, 2009). He is a significant participant not only in the British debate, insisting on avoiding alarmist language when communicating about climate change and options for climate policy.

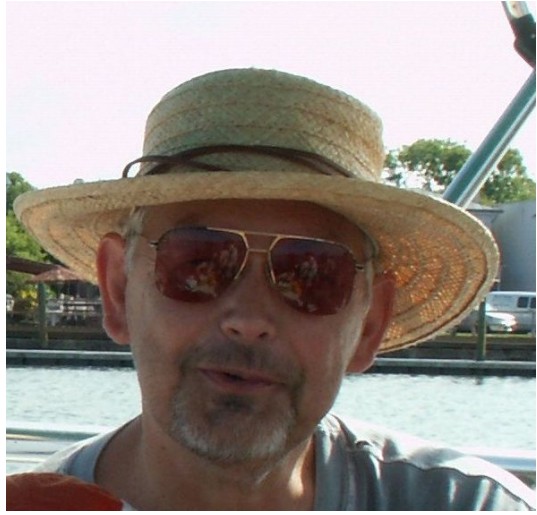

**Figure 13. Thomas J. Crowley**, a US geologist dealing with paleoclimatology was for most of his life located in the US, but in the 2000s he moved with his wife Gabriele Hegerl, herself an eminent scientist, to Edinburgh, UK.


about the state of science, its developments in recent years, and the perspectives for the future. See http://hvonstorch.de/klima/
Media/interviews/AS/all.pdf

## 3.4 CLISAP interviews

Members of the Hamburg Center of Excellence "Integrated Climate Analysis and Prediction" (CLISAP) were interviewed by
Mike Schäfer and Hans von Storch about every month from April 2011 until August 2012. The idea was to illustrate to the
members of the Center the spectrum of disciplines, the variety of topics and people (from professors to PhD students) in the
Center.

The questions dealt mostly with the career of the interviewees, their expectations for their future scientific achievements, the
role of science within society, the degree of politicization of climates science, or what would constitute "good science".

The interviews can be read and be downloaded from http://hvonstorch.de/klima/interviews.htm#clisap.

## 4 Epilogue

This essay presents an attempt to understand climate research and the relationship between climate research and public in a
historical context.

The climate in historical times offers a collection of reactions of people and societies on weather events, weather processes
and climate fluctuations and change. These perceptions and interpretations depended on the evaluation by the respective au-
thorities as Church or science, which in turn are not free from cultural constructions and "Zeitgeist". The historical dimension
therefore offers climate science an excellent possibility of self-reflection; I think this community needs that self-reflection,
and the interviews presented in this article are a manifestation of both, self-reflection but also the reign of cultural constructed
knowledge, and the Zeitgeist. One should also be aware that scientific claims which are consistent with historically pegged
ideas assert themselves particularly well in the "knowledge market" (von Storch, 2023). In this respect, the knowledge about
the historically grown ideas is for the social dealing with climate and climate policy more than useful.

Of course, there is another role of historical research in climate research - the reconstruction of past events, states and changes
during historical times, when human archives are available (e.g., Brázdil et al. (2005); Niedźwiedź et al. (2015); White et al.
(2018); Pfister and Wanner (2021)). Such data may not only serve to directly reconstruct past states, but also as a reference
for reconstructions with proxy-data, or to estimate the range of extreme values, such as, among many others, flooding in the
German Ahr area (Roggenkamp, 2022) or storm surges in the southern Baltic Sea (Wagner et al., 2016) or the Southern North
Sea (de Kraker, 1999).

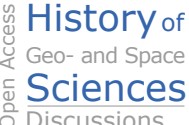

*Acknowledgements.* A significant part of the analysis presented in this article is based on joint work with Nico Stehr, which is documented in their joint anthology (Stehr and von Storch, 2023). The unique and significant long-years work of surveying international climate science scholars by Dennis Bray cannot be underestimated.

Thanks to Heinz Wanner, Sjoukje Philip, Mike Hulme, Jürgen Sündermann, Dennis Bray, Anders Omstedt, George Kiladis, and Hartmut
Heinrich for helping with checking the details of section 3.2. Christian Pfister added some useful results to the full manscript.

*Data availability.* Links to the code books, data, and reports for the surveys (section 3.1) are provided by the project-log of
https://www.researchgate.net/project/The-Bray-and-von-Storch-surveys-among-international-climate-scientists
The photos are from the personal foto archive of Hans von Storch, who is holding a copyright of them.

*Video supplement.*
- A summary on climate science as a social process by Hans von Storch: https://youtu.be/x5E4xEGovrc
- A statement by Anders Omstedt: https://youtu.be/R50cIMcHI6Q



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
