# Peer review of "The role of history in and for climate science Social context and oral accounts"

_History of Geo- and Space Sciences, 2023_

## Referee Comment (RC1)

Confidential and Anonymous evaluation of
Hans von Storch, "The role of history in and for climate science: Social context and oral
accounts," manuscript hgss-2023-6

Hans von Storch is a distinguished climate scientist and commentator on history and social
process. He has interviewed a number of scientists in this field and, in this manuscript, provides
links to these interviews along with very brief comments on changes in social context as climate
policy rose to prominence.

The strengths of the manuscript lie in the access it provides to archived interviews of prominent
climate scientists and to surveys (with Dennis Bray) of professionals in the field. Also of great
value is the reminder that climate science has invested too little in the history of ideas.

Beyond the history of ideas, social and cultural history matter, perhaps more. Humans are, and
always have been, embedded in the climate system. Climate change is not at all new. Elite and
popular ideas, concerns, accommodations, and interventions are woven into the fabric of human
history, from the Pleistocene to the "Anthropocene." It is true that recent environmental
problems have been brought to public notice by scientists and engineers, but the perennial
problems, exacerbated now, belong to us all.

First of all, the author should consider a new title for the piece. "The role of history in and for
climate science: Social context and oral accounts," promises, but does not explain the role of
history. Few, if any historians are cited, and there are no substantial examples or case studies. An
outline of the changing Zeitgeist illustrated by scientific and societal turning points would be
quite illuminating.

This could be accomplished by summarizing the findings of the interviews and surveys
conducted with Dennis Bray, and also summarizing, with analysis, the interviews conducted by
Hans von Storch. Perhaps the focus could be what the interviewees said about professional vs.
popular approaches to climate and climate change. Such interpretive material from the author
would go a long way toward helping the reader "understand climate research and the relationship
between climate research and public [apprehensions?] in a historical context."

I then recommend reworking the existing section 3 by greatly condensing it into a list of
reference links to the interviews.

Hans von Storch has recently completed a forthcoming book with his colleague, sociologist Nico
Stehr, titled *Science in Society: Climate Change and Climate Policies* (World Scientific, April
2023), https://doi.org/10.1142/q0399. Here the authors address the subject of "moving science
into society" in the face of economic, political, and cultural constraints.

Since science is a social activity (not a one-way street) that is practiced by scientists (and groups
of scientists) with economic, political, and cultural interests, this task is not a straightforward or
simple. In fact, over the past three decades, climate science — as practiced by professionally
trained climatologists and oceanographers — has taken a back seat to widespread popular
anxieties about climate change — portrayed as unprecedented and existential emergencies by

pundits and politicians. Perhaps a summary of results from this book would strengthen the article.

In my opinion, this hgss manuscript would be greatly improved by attention to these themes.